# Heat Waves Amplify the Urban Canopy Heat Island in Brno, Czechia

**Zdeněk Janků** [1,2,*] **and Petr Dobrovolný** [1,2]

1   Department of Geography, Faculty of Science, Masaryk University, 611 37 Brno, Czech Republic
2   Global Change Research Institute, Czech Academy of Sciences, 603 00 Brno, Czech Republic
*   Correspondence: janku.z@czechglobe.cz or 501018@mail.muni.cz

**Abstract:** This study used homogenised mean, maximum, and minimum daily temperatures from 12 stations located in Brno, Czechia, during the 2011–2020 period to analyse heat waves (HW) and their impact on the canopy urban heat island (UHI). HWs were recognized as at least three consecutive days with $T_x \geq 30\,°C$ and urban–rural and intra-urban differences in their measures were analysed. To express the HWs contribution to UHI, we calculated the UHI intensities (UHII) separately during and outside of HWs to determine the heat magnitude (HM). Our results show that all HW measures are significantly higher in urban areas. UHII is mostly positive, on average 0.65 °C; however, day-time UHII is clearly greater (1.93 °C). Furthermore, day-time UHII is amplified during HWs, since HM is on average almost 0.5 °C and in LCZ 2 it is even 0.9 °C. Land use parameters correlate well with UHII and HM at night, but not during the day, indicating that other factors can affect the air temperature extremity. Considering a long-term context, the air temperature extremity has been significantly increasing recently in the region, together with a higher frequency of circulation types that favour the occurrence of HWs, and the last decade mainly contributed to this increase.

**Keywords:** mean, maximum, and minimum daily temperatures; heat waves; urban heat island; heat magnitude; Brno; Czechia

## 1. Introduction

The global human population has grown rapidly in urban areas since the middle of the 20th century, from 30% in 1950 to 55% in 2018 and by 2050, the number of the world's population living in urban environments is projected to further increase to 68% [1]. In Europe, the level of urbanization is recently even higher, nearly 75%, with expectations to exceed 80% in the 21st century [2]. Furthermore, the average global surface temperature in the first two decades of the 21st century was approximately 1 °C higher than in the 1850–1900 period, and future projections indicate further global warming even under the low greenhouse gas emissions (GHG) scenario [3]. Current climate change is even more intensified by human activities concentrated mainly in urban areas [4]. Therefore, global climate change, together with the growth rate in urbanization, has a negative impact on the quality of life in the urban environment due to the more frequent occurrence of urban heat load situations [5].

The most studied urban climate phenomenon is the urban heat island (UHI), which is defined by higher air temperatures in urban areas compared to their rural counterparts [6]. UHI can be classified into several types and their definitions, main processes, data collection methods, and impacts can be found elsewhere [7–9]. The urban population is arguably the most affected by atmospheric canopy UHI, and its negative consequences can be significantly intensified during summer heat waves [10].

Heat waves (HW) are the most pronounced extreme air temperature events in the context of global climate change due to their negative impacts on people's health, and recent studies show that research dealing with heat waves is a globally widespread and highly

dynamic topic [11]. Mora et al. [12] estimated that about 30% of the global population is exposed to excessive heat load, with this percentage increase in the range from 48% to 74% by 2100, depending on the GHG scenario. In Europe, HWs are recognized as one of the most worrying and deadliest natural disasters [13], since these heat-related events could cause thermal stress that could lead to deterioration of public health [14,15] or even a higher number of deaths [16,17], particularly in risk groups of the population, such as the elderly, those with low incomes, or with heart disease [18–20].

Society vulnerability issues associated with HWs have received more attention in recent decades since the extremity of air temperature conditions has increased [21–23]. In the 21st century, Europe has experienced several intense HW periods in which tens of thousands of people prematurely died, for example, in parts of western Europe in 2003 [24,25], eastern Europe in 2010 [26,27], and central Europe in 2015 [28,29]. Furthermore, according to future climate projections, the intensity, frequency, duration, and severity of HWs will likely increase until the end of the 21st century [30,31], as well as the number of tropical nights, summer days, and hot days [32–34]. However, the rate of this increase will be strongly influenced by future climate policies [35].

At this point, it is clear that warming due to both global climate change and the UHI effect negatively affects the urban environment. However, the concept of UHI is very complex, and the rate of increase in air temperature conditions due to global warming and the growth of UHI strength may not be in the same mechanism [36]. Although the intensity of UHI increases not only by factors driven by the urban environment but also by macroclimatic dynamics, the relationship between UHI and heat-related events is still uncertain [37]. Many studies, particularly in major mid-latitude cities where the most intensive UHIs have been observed, conclude that UHI intensity is effectively amplified during HWs [38–43]. In contrast, some studies show that synergy is not significant and other factors that influence urban–rural differences are more important drivers of this variability [44–46] and some studies even propose that the intensity of UHI decreases during HWs [47–49]. However, most studies report that HWs favour more intensified outdoor thermal discomfort, particularly in urban areas due to the UHI effect. As a result, the combined effect of both climate change and UHI increases extreme heat exposure and has negative consequences for the quality of life in the urban environment [50]. Consequently, a better understanding of the past, current and possible future of urban climate is required to improve city planning management and policy with potential adaptation and mitigation strategies to avoid or ease the impact of heat load situations.

Brno as a midsized central European city with a densely built-up city centre and location in complex terrain favours the formation of the UHI. Brno has a long tradition of air temperature measurements, both fixed and mobile, which have been used in research many times before [51,52]. Some studies also explore surface UHI with thermal and satellite images [53,54], as well as the climate modelling approach to simulate current and future climate conditions [55,56]. However, most of these studies focused only on UHI patterns and, so far, no attention has been paid to the possible synergy between UHI and HWs. Furthermore, neither of these studies elaborated on detailed meteorological observation data from the most recent extremely warm decade (2011–2020).

In this paper, special-purpose temperature measurements are used to analyse several HW measures and their impact on the UHI intensity. This analysis would provide answers to the following questions: (1) What are the changes in the characteristics of heat waves in the conditions of the urban canopy heat island? (2) To what extent do heat waves increase the intensity of the urban canopy heat island? (3) What are the factors that modify the intensity of the urban canopy heat island during heat waves?

## 2. Materials and Methods

### 2.1. Study Area

Brno (49.2° N, 16.5° E) is situated in the South Moravian Region (Figure 1) and is the second largest city in Czechia (CZ) with a cadastral area of 390 km$^2$ and a population of

382,405 inhabitants. The study area is located at the confluence of two rivers Svratka and Svitava in a complex terrain basin with a mean elevation of 259 m and an altitude range from lower parts set in the south and east with a minimum of 190 m to higher parts in the north and west with a maximum of 497 m within the cadastral area of Brno. More than half of this area is covered by agricultural and arable land (54%) and natural areas including forest land and urban greenery even near the city centre also have a significant share (32%). In the north-western part close to the built-up areas, there is also a large water reservoir Brno with a surface area of 2.59 km$^2$. However, the increase in industrial areas and built-up areas, particularly in agricultural land uptake, is evident in the 21st century.

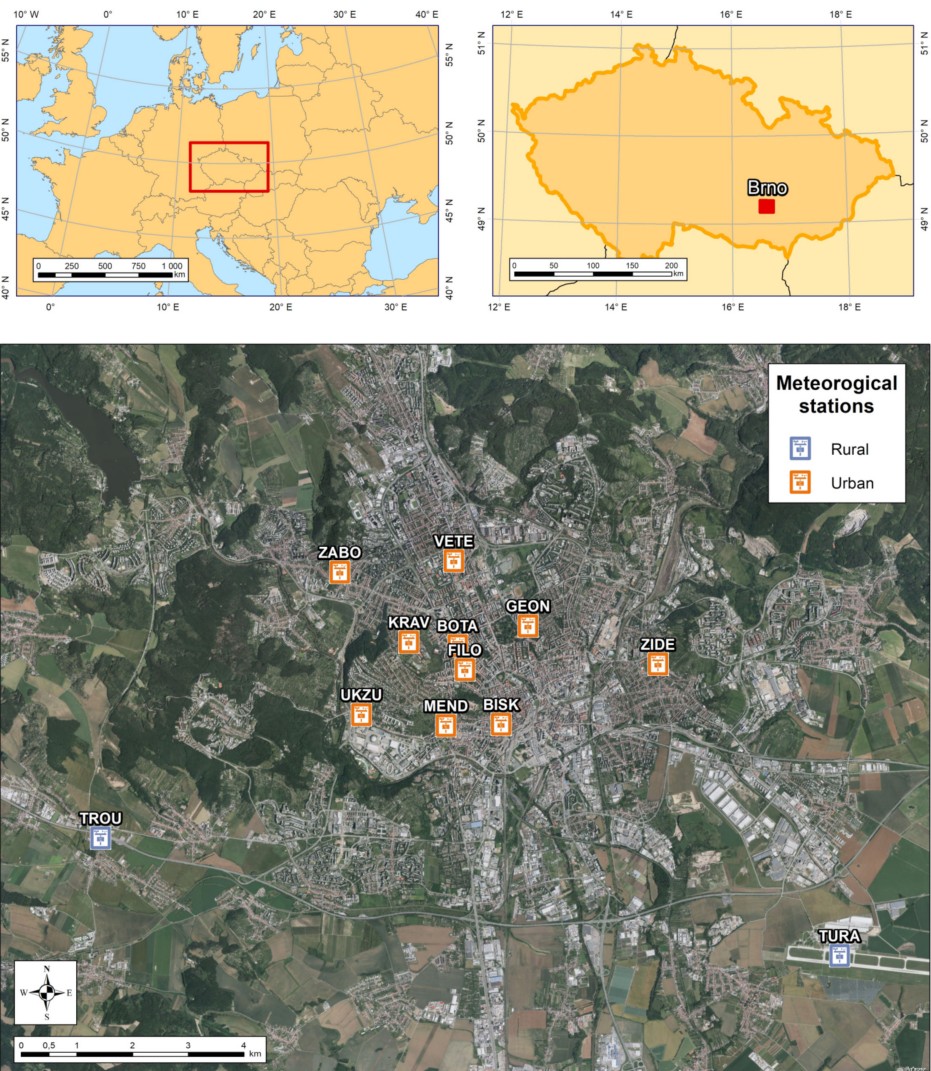

**Figure 1.** The location of Brno within central Europe with the positions of selected meteorological stations within the ortofoto image of Brno and its surroundings [57].

From a climatological perspective, Brno is located in the moderate Cfb climate zone according to the Köppen–Geiger climate classification. Furthermore, Brno is situated in one of the warmest and driest regions in Czechia. In the 1961–2020 period, the mean annual air temperature was 9.4 °C, with a maximum in July (19.6 °C) and a minimum in January (−1.7 °C), and the mean annual precipitation was approximately 500 mm. Nevertheless, meteorological data from Brno, Tuřany station have shown a significant increase in air temperature conditions during recent years, shifting the value of the mean annual air temperature from 8.7 °C in the 1961–1990 period to 10 °C in the 1991–2020 period. Furthermore, the warmest air temperature conditions were recorded in the last decade

(2011–2020), with the mean annual air temperature reaching 10.9 °C. The changes in other meteorological variables were not that significant.

*2.2. Meteorological Data*

The measurement network used in this study consisted of altogether twelve stations (Figure 1 and Table 1) and has been compiled from two sources. The first source represents daily air temperature data ($T_a$, $T_x$, and $T_n$) from three stations of the Czech Hydrometeorological Institute (CHMI) compiled from 10-min measurements. The ZABO station is located in the north-western part of the Brno built-up area with UHI influence. In contrast, the TROU and TURA stations are located south of the city, mainly surrounded by arable lands with a great share of vegetation, as a result representing the rural climate. The TURA site, located at the airport, provides the longest measurement period (1961–2020) and was used to characterize the long-term variability of HW measures in the Brno area and to compare with the general tendencies of HWs in Czechia. The second source of data was 10-min measurements at nine automatic meteorological stations (AMS) of the special-purpose network that covered the last decade (2011–2020). These stations are located within the built-up areas (Figure 1; Table 1).

**Table 1.** Meteorological stations with several descriptive characteristics: local climate zones (LCZ), altitude (ALT), sky-view factor (SVF), building surface fraction (BSF), and normalized difference vegetation index (NDVI). See the text for further explanation.

| NO. | CODE | TYPE | REGIME | LCZ | ALT (m) | SVF | BSF (%) | NDVI |
|-----|------|------|--------|-----|---------|-----|---------|------|
| 1 | BISK | Urban | Special | 2 | 245 | 0.50 | 35.9 | 0.219 |
| 2 | BOTA | Urban | Special | 2 | 242 | 0.72 | 32.7 | 0.290 |
| 3 | FILO | Urban | Special | 5 | 234 | 0.47 | 30.0 | 0.352 |
| 4 | GEON | Urban | Special | B | 225 | 0.81 | 16.1 | 0.514 |
| 5 | KRAV | Urban | Special | 9 | 298 | 0.97 | 14.9 | 0.507 |
| 6 | MEND | Urban | Special | 5 | 206 | 0.65 | 18.6 | 0.413 |
| 7 | TROU | Rural | Standard | D | 278 | 0.84 | 5.1 | 0.430 |
| 8 | TURA | Rural | Standard | D | 241 | 0.98 | 4.1 | 0.429 |
| 9 | UKZU | Urban | Special | 5 | 214 | 0.80 | 16.2 | 0.366 |
| 10 | VETE | Urban | Special | 8 | 237 | 0.70 | 25.9 | 0.394 |
| 11 | ZABO | Urban | Standard | 5 | 236 | 0.69 | 18.1 | 0.458 |
| 12 | ZIDE | Urban | Special | 9 | 216 | 0.83 | 16.5 | 0.486 |

*2.3. Land Use and Surface Characteristics*

Urban–rural and intra-urban air temperature differences are generally made by environmental and anthropogenic factors that influence spatio-temporal variability of air temperature distribution. However, the role of these factors is highly dependent on the character of the city, e.g., geographical location, area and population size or urban structure [58]. Therefore, in this study, land use and surface characteristics are used in the analysis of explanatory factors. All descriptive characteristics are calculated for the specific location of the selected meteorological stations (Table 1).

In this study, the land use classification in the form of the local climate zones (LCZ; Figure 2a) was used to provide high-resolution (100 × 100 m) information on the character of the surface with 14 land use classes in total for the city of Brno and its rural surroundings [59]. Most of the selected stations are located within the built-up areas represented by LCZ classes 1–10. However, one of the urban stations (GEON) and both rural stations (TROU and TURA) are located in the land cover areas mostly surrounded by a natural environment and are represented by LCZ classes A–G.

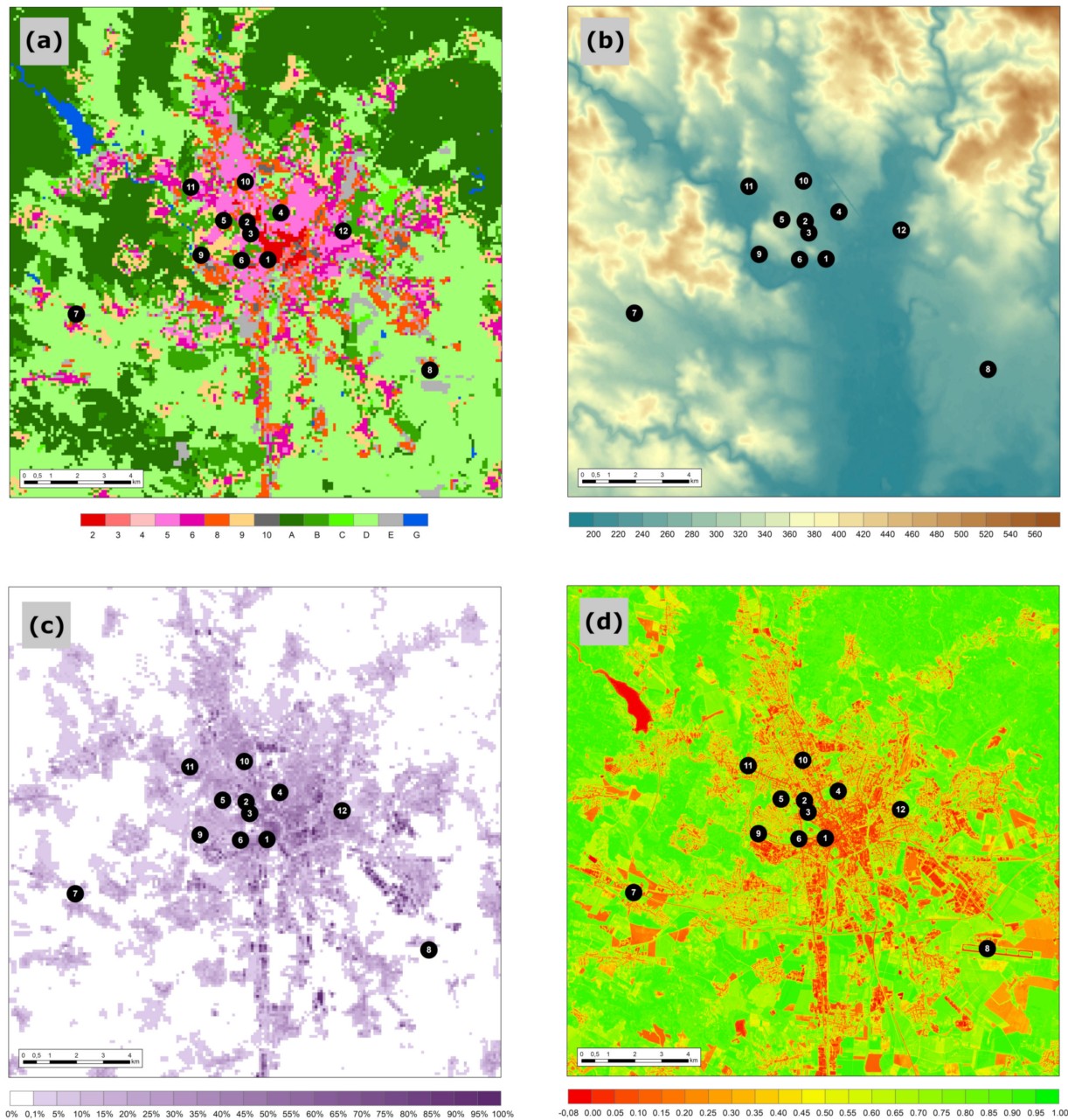

**Figure 2.** Spatial distribution of (**a**) Urban Atlas land use classes, (**b**) altitude, (**c**) building density, and (**d**) vegetation in Brno and its close surroundings.

The altitude variable (ALT; Figure 2b) representing the topography of the study area is retrieved from the EU-DEM v1.0 grid model of the EEA with a resolution of 25 × 25 m. The study area ranges from 182 m above sea level in the southern part to 592 m in the northern part. However, all meteorological stations have elevations in the range of 200 to 300 m above sea level. The sky-view variable (SVF; Table 1) describes the level of sky obscured by surrounding barriers, and this was measured geodetically only at the selected sites. In the study area, the SVF variable ranges from 0.47 to 0.98 with the highest values primarily in rural sites or in sites far from the city centre. The building surface fraction layer (BSF; Figure 2c), with high-resolution 100 × 100 m, represents the percentage of area covered with an artificial building structure, and the BSF parameter was calculated for selected stations using buffer analysis. The BSF variable fluctuates from 4.1% to 35.2% in a radius of 300 m from the stations. Generally, building cover areas can be found near the city centre or industrial zones. Finally, the normalized difference vegetation index (NDVI; Figure 2d) is

used to quantify the amount of vegetation on the surface. The NDVI values were calculated for selected stations using buffer analysis on the raster layer with the high-resolution $10 \times 10$ m obtained from the WEkEO portal of the EEA. The NDVI variable ranges from 0.219 to 0.514 in a radius of 300 m from the stations and generally reaches higher numbers at sites where the BSF values are lower.

*2.4. Methods*

2.4.1. Homogenisation

The study of air temperature in a highly heterogeneous urban environment requires high-quality and spatio-temporally detailed data. Air temperature data measured by special-purpose stations can provide valuable information on this issue. However, such measurements are unlikely to be completely accurate. Therefore, homogenisation and in-filling of the missing data are required to effectively reduce these inhomogeneities [60]. In this study, the R package Climatol [61] was applied for quality control, homogenisation, and in-filling of missing data in a set of daily air temperature variables. After removing obviously incorrect records, the homogenisation process calculated the provisional means and standard deviations of the time series with the available data to fill in the missing records. After that, Alexandersson's Standard Normal Homogeneity Test (SNHT) was applied for the detection of inhomogeneities [62].

2.4.2. Statistical Analysis

Homogenised daily temperatures were used for the selection of HWs at individual stations. There are several definitions of HW in different regions using both air temperature and apparent temperature. For the purposes of this study, an HW is defined as a period of at least three consecutive days with $T_x \geq 30$ °C, a general definition used many times before in this particular region [23,51,63]. After the HWs are recognized, a set of basic descriptive characteristics is applied to determine their quantity, duration, intensity, and severity. The following characteristics of HWs are analysed: an annual number of HWs (QHW), an HW length in days (DHW), a sum of $T_x$ during HW (IHW), and an average $T_x$ in HW (SHW). To characterize these HW measures at individual stations, their box-plots (median, average, first and third quartile, minimum and maximum) were calculated.

Urban–rural variability in air temperature conditions due to UHI fluctuates depending on weather patterns, day-time, land use, location, and character of the city. Here, we use the urban heat island intensity (UHII) calculated as the difference between the air temperature conditions at urban and rural stations [37]. Due to relatively heterogeneous rural conditions in the city neighbourhood (Figure 2a), rural conditions are calculated as a mean temperature measured at two rural stations (TROU and TURA). Besides UHII calculated from the daily mean air temperatures ($T_a$), we also analysed UHII calculated from daily minima ($T_n$) and daily maxima ($T_x$) that describe night-time and day-time UHII, respectively.

In addition, we calculate the heat magnitude (HM) to evaluate the degree of heat load modification in urban areas during HWs [64]. HM is quantified as the difference between the average UHII during HW days ($UHII_{HW}$) and the average UHII during non-HW days ($UHII_{NHW}$). Due to the dominant HW occurrence in summer months, HM is analysed for the JJA season and for $T_a$, $T_x$, and $T_n$, respectively. The significance of HM was evaluated using the Wilcoxon rank-sum test comparing $UHII_{HW}$ and $UHII_{NHW}$.

The density curves for $T_a$, $T_x$, and $T_n$ constructed for HWs and non-HWs days allowed the evaluation of possible differences in the air temperature distribution. The density estimates were compared both graphically and formally with the bootstrapping method implemented in the sm R package [65]. This method allows the testing of equality between two density curves producing a reference band and the corresponding *p*-value. The course of the density curves outside the reference band indicates a significant difference in the air temperature distribution in HWs and non-HWs days.

Spearman's rank correlation coefficient was applied to quantify the relationship between land use and surface characteristics and the selected indicators of extreme air tem-

perature conditions (HW measures, UHII, and HM) at individual sites. Descriptive characteristics summarized in Table 1 were used as explanatory variables.

As the HW measures for the TURA airport station have been available since 1961 we analysed their linear trends. The significance of the monotonic trend was evaluated using the non-parametric Mann–Kendall test [66], and the Theil–Sen method was used for the estimation of the trend parameters [67]. Linear trends were also smoothed by the 5-year moving average.

## 3. Results

### 3.1. Urban–Rural and Intra-Urban Differences in the HW Measures

Generally, significantly higher numbers of HW characteristics during the 2011–2020 period were found in urban stations compared to rural sites. Urban–rural differences can be observed in all HW measures, and they are clearly expressed in QHW and SHW, although not so evident in IHW and DHW (Figure 3). However, both IHW and DHW are highly dependent on the number of days in the HW, and thus their distribution is asymmetric since the majority of HWs have a short-term duration of 3 or 4 days (Figure 4). Therefore, the distribution of values is quite similar, but the frequency is not. Urban stations have on average a higher number of both short and long HWs. Furthermore, the duration and intensity in the case of individual HWs during the same period are generally higher at urban stations. Therefore, HWs are more frequent, longer, more intense, and more severe in urban areas of Brno.

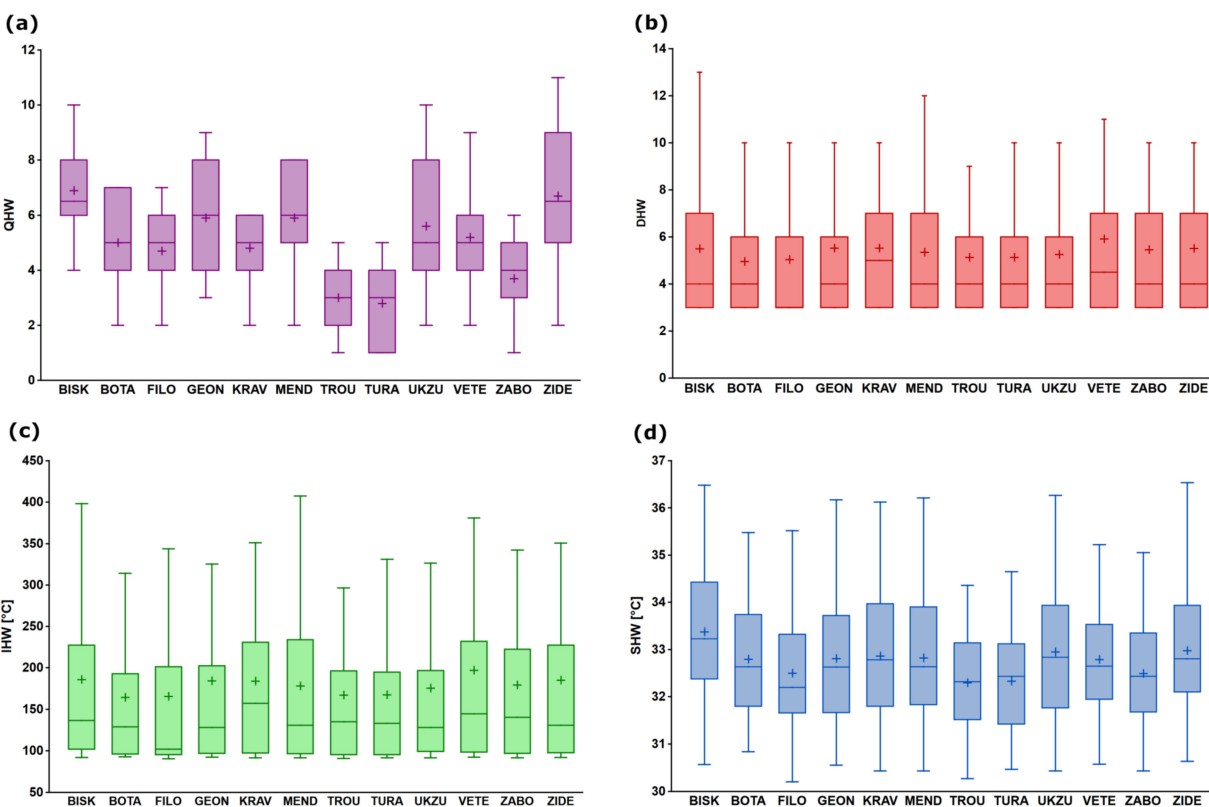

**Figure 3.** Box-and-whisker plots (median, average, first and third quartile, minimum, and maximum) characterizing (**a**) an average annual number of HW days (QHW), (**b**) the length of HW in days (DHW), (**c**) a sum of $T_x$ during HW (IHW), and (**d**) a mean $T_x$ in HW (SHW) for HWs recorded at selected meteorological stations in Brno in the 2011–2020 period.

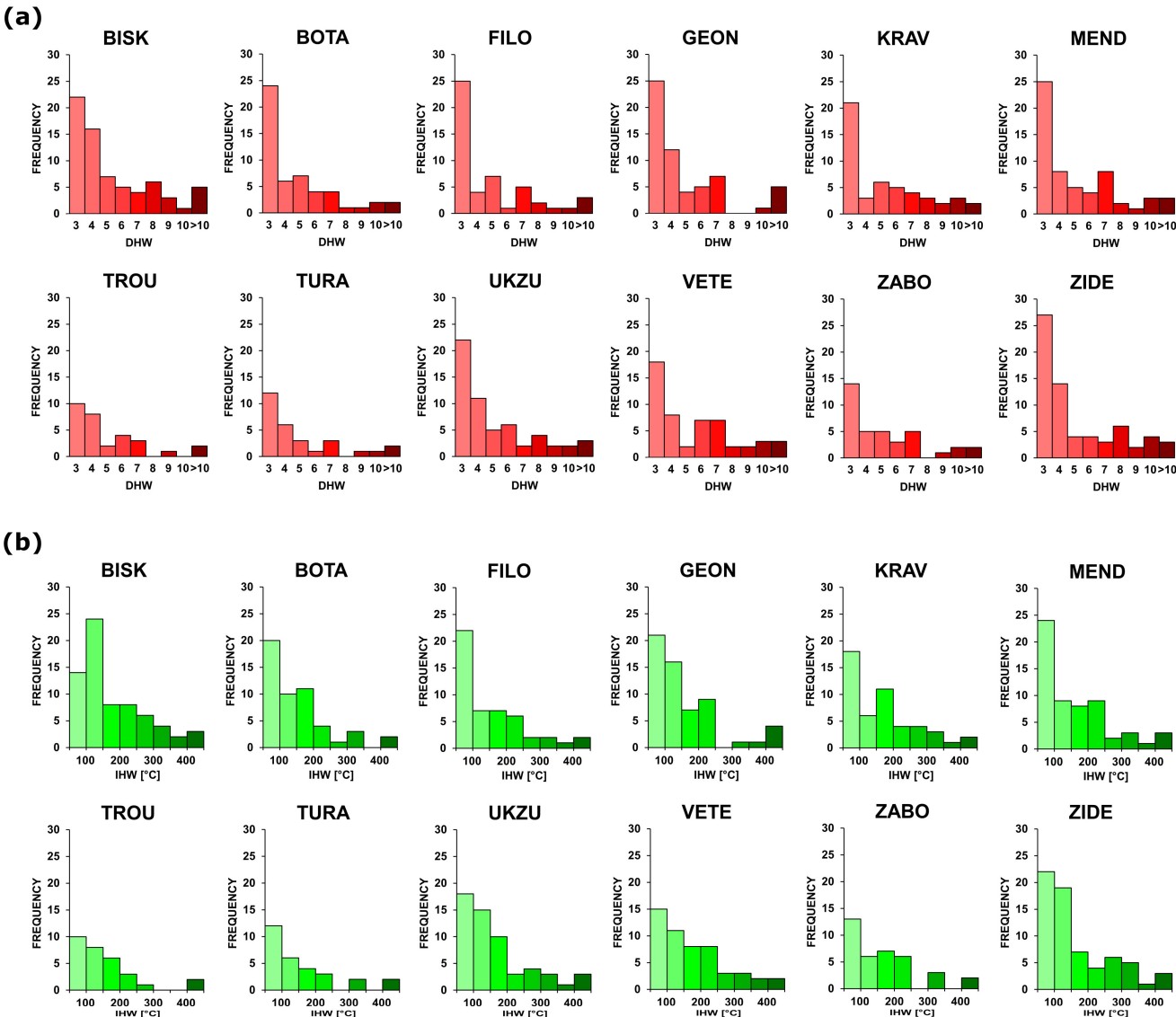

**Figure 4.** Distribution of (**a**) the length of HW in days (DHW), and (**b**) a sum of $T_x$ during HW (IHW) for HWs recorded at selected meteorological stations in Brno in the 2011–2020 period.

The HW measures are quite similar in two rural stations, TROU and TURA (LCZ D) (Figure 3). On average, three HWs occurred there each year with an average duration of about five days (Figure 3a,b). In total, 30 HWs were recorded in TROU and 28 HWs in TURA during the study period, fluctuating from a single HW a year to five. Long HWs (at least 10 days) occurred in hot summers in 2015 and 2018 with the longest lasting 15 days from 27 July to 10 August 2018. IHW and SHW at TROU and TURA sites also took on similar values with the sum of $T_x$ in a range of 167–168 °C and the mean $T_x$ of 32.3 °C on average in the 2011–2020 period (Figure 3c,d).

While HW characteristics have similar values in the rural neighbourhood of Brno, they varied widely at urban stations, especially in the cases of QHW, IHW, and SHW. The total number of recorded HWs typically fluctuates between 4–6 HWs per year, although at some stations (e.g., BISK and ZIDE, LCZ 2 and LCZ 9, respectively) it has more than doubled compared to the rural station, exceeding 6 HWs per year (Figure 3a). If the typical length of HWs ranged between 5 and 6 days at all selected stations (Figure 3b), the overall distribution of HWs length changed significantly, as both shorter and longer HWs generally occur more often in urban stations. The total number of long HWs (at least 10 days) has increased from 2–3 in rural areas to 6–7 in some urban sites (the same as in QHW (Figure 4a).

The absolute longest HW was recorded at the ZIDE station (LCZ 9) between 23 July and 24 August 2018, lasting 33 days and had an IHW value of nearly 1200 °C. This HW also reached similar measures at other urban stations (e.g., GEON and VETE, LCZ B and LCZ 8, respectively). Moreover, typical IHW values at urban stations increased by 10–15% in some urban stations compared to values at rural ones (Figure 3c). As in the DHW measure, there was no significant value range in HW intensity on average, but the overall distribution of IHW values shows more considerable intra- urban differences (Figure 4b). Finally, typical SHW values ranged between 32 °C and 34 °C at all urban stations (Figure 3d), although for individual HWs they differed significantly. The most severe HW was recorded from 3 August to 16 August 2015 with an average $T_x$ exceeding 36 °C at several urban stations and even reaching 38 °C at the BISK site (LCZ 2), approximately 4 °C higher than at the rural site TURA (LCZ D).

### 3.2. Urban Heat Island Intensity and the Contribution of HWs

To express the contribution of HWs to urban heat islands, the UHII values of each urban station were calculated for all days during the JJA season of the 2011–2020 period. Subsequently, the characteristic UHII values during the HW days and the outside HW days ($UHII_{HW}$ and $UHII_{NHW}$) were compared to estimate the HM values representing HWs contribution to UHII. Since UHII creates a typical daily course with a maximum intensity generally during the night hours in central European cities, the UHII and HM values were analysed separately for $T_a$, $T_n$, and $T_x$ characterizing this daily UHII evolution at individual urban stations (Figure 5 and Table 2).

**Table 2.** Average UHII values in days without recorded HW ($UHII_{NHW}$) or with recorded HW ($UHII_{HW}$), and heat magnitude (HM) values within the urban stations of Brno during the JJA season in the 2011–2020 period. An asterisk (*) indicates statistically significant values ($p < 0.05$) of UHII and HM.

| STATIONS | $T_a$ | | | $T_x$ | | | $T_n$ | | |
|---|---|---|---|---|---|---|---|---|---|
| | $UHII_{NHW}$ | $UHII_{HW}$ | HM | $UHII_{NHW}$ | $UHII_{HW}$ | HM | $UHII_{NHW}$ | $UHII_{HW}$ | HM |
| BISK | 1.30 * | 1.63 * | 0.33 * | 2.60 * | 3.49 * | 0.89 * | 1.29 * | 1.80 * | 0.51 * |
| BOTA | 0.64 * | 0.66 * | 0.02 | 1.36 * | 1.61 * | 0.25 * | 0.62 * | 0.84 * | 0.23 * |
| FILO | 0.68 * | 0.80 * | 0.12 * | 1.08 * | 1.43 * | 0.35 * | 0.89 * | 1.09 * | 0.20 * |
| GEON | 0.28 * | 0.26 * | −0.02 | 2.07 * | 2.54 * | 0.47 * | −0.02 | 0.10 * | 0.12 * |
| KRAV | −0.22 * | 0.03 | 0.25 * | 1.28 * | 1.93 * | 0.65 * | −0.70 * | −0.59 * | 0.11 |
| MEND | 1.09 * | 1.03 * | −0.06 | 1.97 * | 2.28 * | 0.31 * | 1.09 * | 1.16 * | 0.06 |
| UKZU | 0.42 * | 0.38 * | −0.04 | 1.70 * | 2.27 * | 0.57 * | 0.22 * | 0.34 * | 0.12 * |
| VETE | 0.83 * | 0.92 * | 0.09 | 1.87 * | 2.26 * | 0.39 * | 0.31 * | 0.49 * | 0.18 * |
| ZABO | 0.68 * | 0.78 * | 0.10 | 0.83 * | 0.93 * | 0.10 * | 0.60 * | 0.64 * | 0.04 |
| ZIDE | 0.44 * | 0.41 * | −0.03 | 2.24 * | 2.96 * | 0.72 * | −0.37 * | −0.30 * | 0.07 |

The UHII values of the urban stations were mostly positive, confirming the UHI formation. The differences between the mean air temperature regimes in urban and rural stations were on average 0.65 °C (Table 2). However, around the time when $T_x$ occurred in the daily course of air temperature, UHII was generally higher, and all urban stations indicate the existence of UHII both during and outside HWs. Furthermore, only in the case of $T_x$, do all urban stations show a statistically significant and positive contribution of HWs to UHII. Although the mean UHII at the time of the maximum daily temperature (mostly in the early afternoon hours) was 1.7 °C during non-HW days, it is on average approximately 0.5 °C higher during HW days. Maximum average temperature excess was found during HWs for the BISK site in the city centre (LCZ2), where HM was on average approximately 0.9 °C.

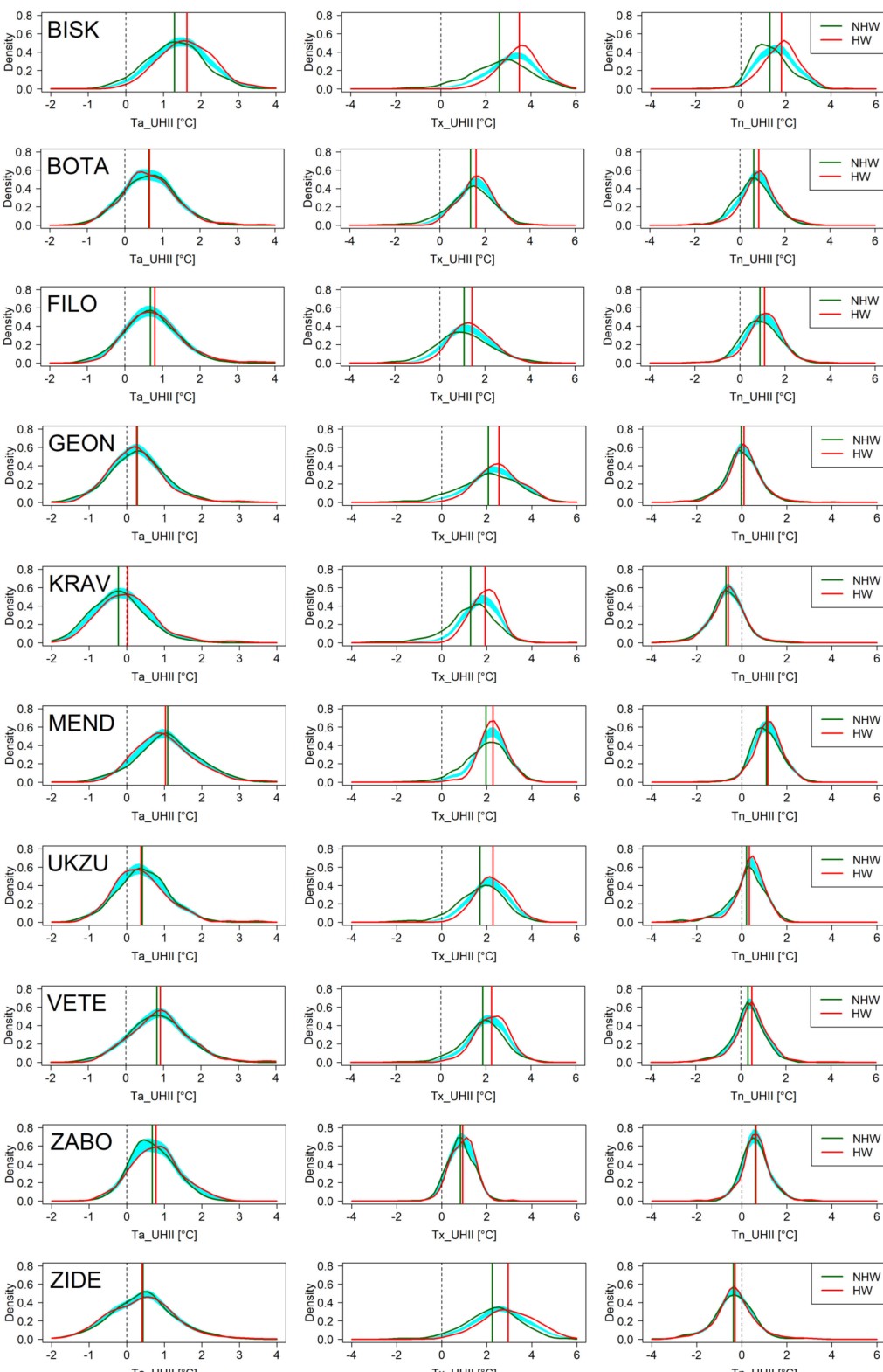

**Figure 5.** Urban heat island intensity (UHII) empirical density curves calculated from HW days (HW) and non-HW days (NHW) during the JJA season of the 2011–2020 period at Brno urban stations. The density curves outside the reference band (blue) indicate a significant difference in the two distributions (and vice versa). Vertical lines are mean UHIIs, and their differences express the heat magnitude (HM).

The results for $T_a$ and $T_n$ are not as unambiguous. UHII was even negative at one station in the case of $T_a$ and at two stations in the case of $T_n$ (Table 2). At the time of the occurrence of $T_n$ (generally at night), significant negative UHII values were recorded at the KRAV and ZIDE sites (both LCZ9) even during HWs. Nevertheless, at the time of $T_n$, the UHII values are mostly positive, with about 0.5 °C on average. Furthermore, the results show that despite the negative UHII values at some urban stations, the HM contribution at night is positive at all stations; at six of them even statistically significant. The smallest enhancement of UHII during HWs was observed for $T_a$ with HM only about 0.1 °C on average. The increase in $T_a$ during HW days was significant only at three stations (BISK, FILO, and KRAV), while HM was found even negative at four stations (GEON, MEND, UKZU, and ZIDE), although with no statistical significance.

Comparing the distribution of temperatures during and outside the HWs through their density function mainly shows larger positive asymmetry (skewness) and kurtosis in the $T_x$ and $T_n$ distribution. There is also a clear shift of $T_x$ to higher values during the HWs (Figure 5). The bootstrapping test proved a significant difference in the $T_x$ distribution during and outside HWs at all stations analysed. However, for $T_n$ such a significant difference was not found at two stations (KRAV and ZABO) and for $T_a$ even at six stations (BOTA, FILO, GEON, UKZU, VETE, and ZIDE). Therefore, the UHI in Brno is often amplified during the HW days, but particularly during the day-time when $T_x$ is likely to occur.

### 3.3. Factors Affecting Differences in the Occurrence of HWs, UHII, and HM

In this section, the characteristics of selected meteorological stations summarized in Table 1 were applied as explanatory variables in the correlation and regression analysis with the aim of quantifying the amount of explained variability in HWs, UHII, and HM (Table 3 and Figure 6). In our analysis, we focus only on QHW and SHW, since the other two HW measures, DHW and IHW, are highly dependent on the number of days in HW and are thus more suited for the evaluation of individual HW. A correlation analysis with the UHII and HM parameters was performed for $T_a$, $T_x$, and $T_n$.

**Table 3.** Spearman's rank correlation coefficients between descriptive characteristics and QHW and SHW, and UHII and HM for $T_a$, $T_x$, and $T_n$ for meteorological data measured at stations within the Brno area during the JJA season of the 2011–2020 period. An asterisk (*) indicates statistically significant correlations ($p < 0.1$).

| MEASURES | QHW | SHW | UHII_$T_a$ | HM_$T_a$ | UHII_$T_x$ | HM_$T_x$ | UHII_$T_n$ | HM_$T_n$ |
|---|---|---|---|---|---|---|---|---|
| **ALT** | −0.43 | −0.25 | −0.01 | 0.87 * | −0.18 | 0.15 | −0.01 | 0.45 |
| **SVF** | −0.36 | −0.17 | −0.87 * | −0.25 | 0.15 | 0.36 | −0.93 * | −0.32 |
| **BSF** | 0.48 | 0.34 | 0.83 * | 0.30 | 0.01 | −0.21 | 0.85 * | 0.64 * |
| **NDVI** | −0.15 | −0.11 | −0.64 * | −0.25 | 0.08 | 0.07 | −0.76 * | −0.76 * |

While all HW measures are negatively correlated with altitude (ALT), sky-view factor (SVF), and amount of vegetation (NDVI), they are positively correlated with the density of built-up areas (BSF) (Figure 6). The influence of BSF on the increase in the number of HW measures is the highest of all land use and surface characteristics. On the contrary, NDVI has the lowest impact on HW measures. However, all the values of the correlation coefficients between the HW measures and the explanatory variables are statistically insignificant. Thus, neither of the selected descriptive characteristics significantly affected the variability in the occurrence and severity of HWs.

The correlation coefficients between descriptive characteristics and UHIIs at urban stations are quite similar in the cases of $T_a$ and $T_n$ (Table 3). No correlation was found with ALT, proving that altitude is not a significant determinator of urban–rural differences. The other three parameters correlate with UHII significantly in the cases of $T_a$ and $T_n$. The highest correlation was found for SVF. UHII demonstrate a tendency to decrease with an increase in SVF and with an increase in NDVI as well, although the decrease rate is lower

than with SVF. In contrast, UHII increases with increasing BSF. This increase is clearly stronger, and therefore areas with a higher density of buildings in Brno are less affected by night-time cooling.

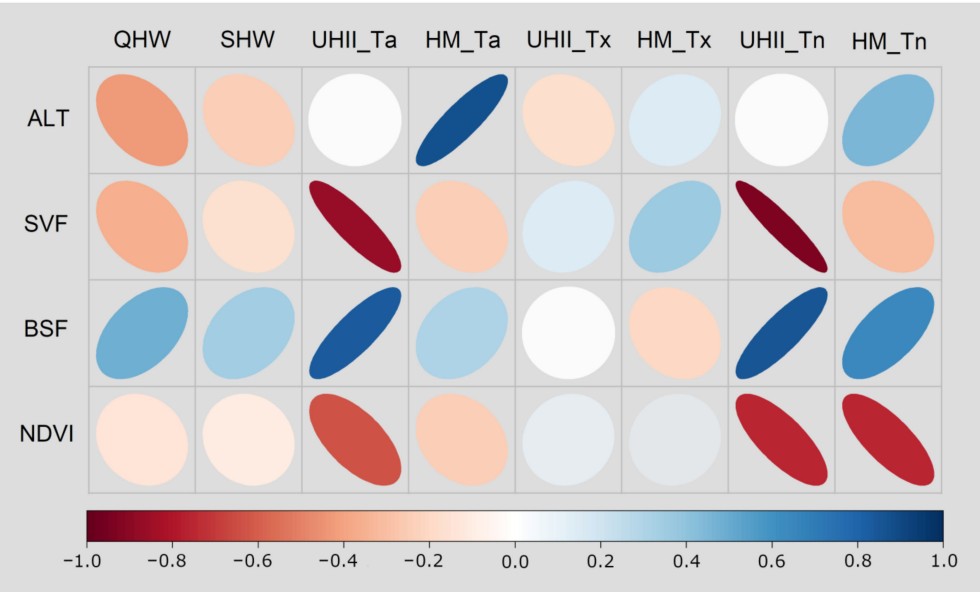

**Figure 6.** Correlation plots between descriptive characteristics and QHW and SHW, and UHII and HM for $T_a$, $T_x$, and $T_n$, based on the Spearman's rank coefficient for meteorological data measured at stations within the Brno area during the JJA season of the 2011–2020 period.

If the UHII derived from $T_a$ increases significantly with the SVF, BSF, and NDVI parameters, this is not the case for the HM of $T_a$. Correlations with HM continue to follow a pattern similar to that in UHII, but the increase or decrease rate is not significant (Figure 6). In contrast, correlations between the HM of $T_n$ and the BSF and NDVI also follow the same pattern, but this relationship is statistically significant. Therefore, the attributes of land use in the close vicinity of the station can considerably affect the amount of added heat load during HWs at night. However, in the case of SVF, the relationship with HM of $T_n$ is not significant. On the contrary to UHII, HM is positively correlated with ALT, for $T_a$ even significantly.

Although the relationships between descriptive characteristics and UHII are often quite similar and even statistically significant in the case of $T_a$ and $T_n$, UHII derived from $T_x$ does not correlate well with these parameters (Figure 6). The strongest negative correlation with UHII was found during the day-time for ALT. However, the correlation is not significant. Furthermore, the correlation with the other three parameters is extremely lower compared to the UHII calculated from $T_a$ and $T_n$. Similarly, all these descriptive characteristics have a low correlation with HM, especially in the case of BSF or NDVI. SVF has the highest correlation with HM derived from $T_x$ of all explanatory variables, but with no statistical significance.

## 4. Discussion

### 4.1. Long-Term Changes in the HW Measures in Brno

This paper deals with HWs and their influence on UHI intensity in Brno during the 2011–2020 period. Consequently, this detailed analysis may be placed in a wider context in time and space. Data from the TURA airport station can be used to compile HW measures from 1961 for trend analysis (Figure 7).

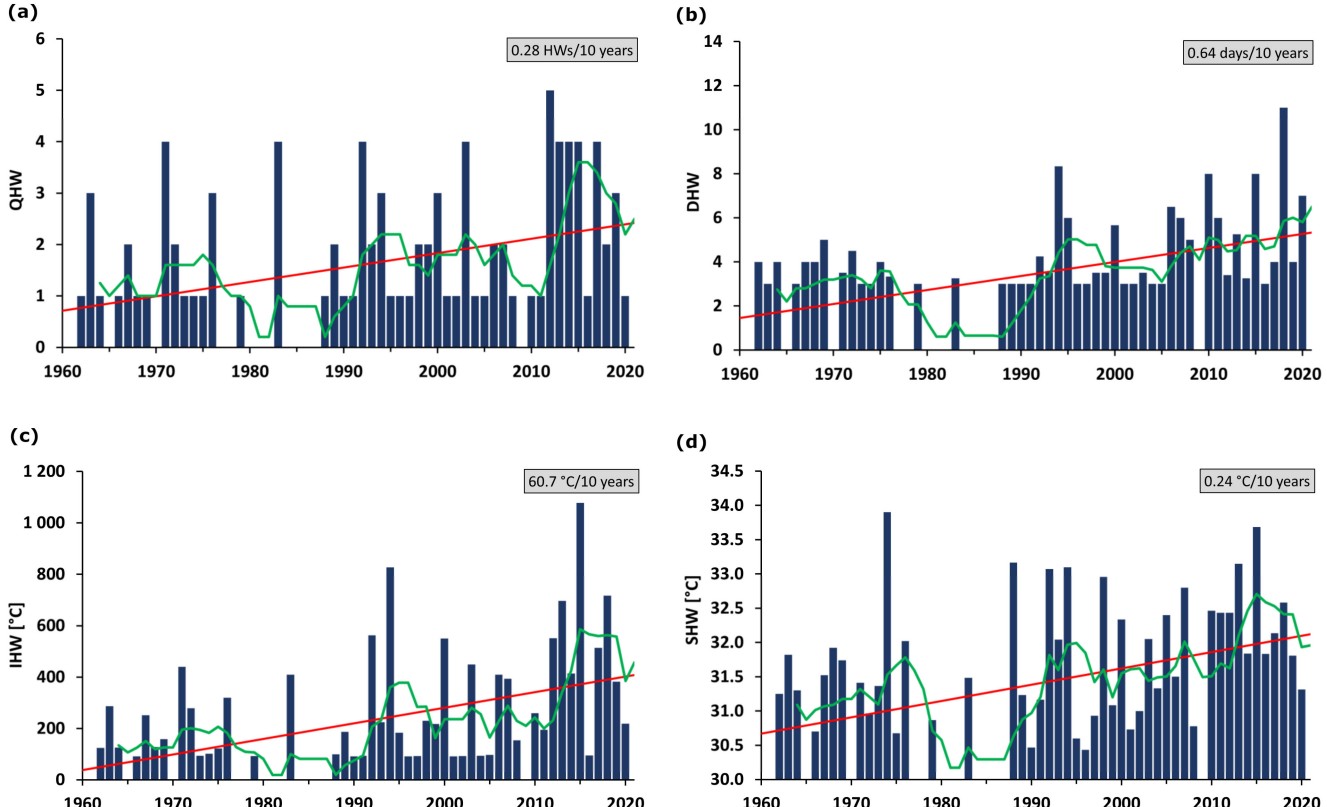

**Figure 7.** Annual numbers of (**a**) HW days (QHW), (**b**) the average length of HW (DHW), (**c**) the sum of $T_x$ during HW (IHW), and (**d**) the mean $T_x$ in HW (SHW) recorded at the TURA station in the 1961–2020 period with a linear trend, a 5-year moving average, and a 10-year increase rate.

There is an obvious increase in HW measures in Brno during the 1961–2020 period, and this increasing trend is highly significant ($p < 0.05$) for all HW measures. However, a closer look at the compiled time series shows that the increase was not monotonic throughout the whole analysed period. The period covering the end of the 1970s and the majority of the 1980s was characterized by clearly lower values of all four HW measures. Nevertheless, since the 1990s, HW measures have increased their number significantly. According to the 5-year moving average, the most intense increase falls on the period at the beginning of the 1990s and 2010s. The mean annual values of all HW measures at the TURA station in the 2011–2020 period were significantly higher ($p < 0.05$) than those of the 1961–1990 reference period. Furthermore, ten of the most intense HWs were recorded at the TURA site over the last 30 years and half of them over the last decade. Thus, this decade mainly contributed to the significant increase in air temperature extremity over the past 60 years in Brno.

The analysis of HWs over the territory of Czechia in the 1961–2020 period was made by Zahradníček et al. [23] with data from 133 stations. HWs were defined in the same way as in this study and station data was compiled into the mean series for the Czech territory. They found a significant increasing trend in the number of HW days reaching on average 1.1 days in 10 years, and in the mean duration of HW 0.56 days in 10 years, while the mean $T_x$ during HW increased by 1.2 °C over 60 years. The growth rates in the HW measures found for the TURA station representing the Brno region are considerably higher (Figure 7). This may be partly explained by the location of the Brno region in one of the warmest regions of the country. However, the TURA station represents the rural environment, and it is clear from our analysis of HWs based on the 2011–2020 period that all HW measures in the urban environment are significantly higher. This means that the growth rates in the urban environment can also be significantly higher in the long-term context.

In addition, not only did HWs become recently more frequent and intense in Europe, but this significant increasing trend is also likely to continue in the coming years of the

21st century [68]. According to the climate projections, the number of HW days is expected to increase significantly in all of Europe, but the greatest increase in HW temperatures is projected particularly in central European cities [31]. In Czechia, the frequency of HWs is predicted to be nearly twice as high in the 2020–2049 period compared to the 1970–1999 period, and this increase will be even higher for the most severe HWs [30]. Additionally, severe HWs can become a regular phenomenon by the end of the 21st century, but their magnitude depends mainly on the GHG scenario.

*4.2. Circulation Types and Their Context with HWs and UHII*

The question is what causes the significant increase in the number of HW measures. Since the average length of HWs (DHW) is quite similar at all stations in Brno, which can be related to the fact that the onset and end of HWs are related to the synoptic situation, it is clear that current atmospheric dynamics and weather conditions are also important. Typical HWs occur during the summer months under stable and persistent anticyclonic conditions in which cooler polar air is cut off and hotter equatorial air affects weather conditions in the blocked region [10]. In Czechia, HWs are likely to occur with the presence of a high-pressure ridge from North Africa to Central Europe, an anticyclone in the central part of the Mediterranean, or an eastward anticyclone [69].

Here, we used the database of circulation types from the 1961–2020 period based on the objective classification centring on the geographic midpoint of Czechia to describe the circulation pattern during HW days [70]. The occurrence of HWs in Brno during the JJA season of the 1961–2020 period was generally recorded during anticyclonic circulation types such as A (15.8%), AE (8.7%), ASE (6,8%) and ASW (5.1%), or during directional circulation types with a southward or eastward direction such as SE (8%), S (7.3%) and E (6.3%). On the contrary, the types of northern cyclonic or directional circulation, such as CN, CNE, CNW, N, and NW, were each recorded only twice during HW days in the 1961–2020 period. The greatest positive differences in the occurrence of circulation types during HW days compared to non-HW days of the JJA season were observed in SE (+4.8%), S (+4.7%), AE (+4.3%), and ASE (+4.2%). Furthermore, most of these synoptic situations increase in number throughout the 1961–2020 period. Therefore, this growth of these circulation types may be one of the reasons why the HWs occurrence has increased in recent years.

In our analysis, we found that UHII in Brno is often amplified during the day-time under the influence of the HW situation, and urban areas are therefore exposed to even more severe and dangerous heat load. This synergy between HWs and UHII can be associated with typical conditions of the anticyclonic weather pattern during HWs. These high-pressure synoptic systems have typical conditions that favour the occurrence of HWs such as a nearly cloudless sky, drier air, and lower wind speed [44,71]. In fact, meteorological data from the TURA airport station can confirm this hypothesis with a lower measured amount of precipitation, relative humidity, and wind speed during HW days compared to non-HW days in the JJA season. For example, low wind speed positively contributes to effectively reducing advective cooling in urban areas compared to their rural counterparts [72]. Furthermore, a higher amount of solar radiation leads to an increased surface temperature providing more stored heat flux and, therefore, changing the urban–rural energy balance [73].

*4.3. Factors Affecting the Synergy between HWs and UHII*

Land use and surface characteristics can affect the HW measures at the site, as well as the UHII and HM values. Since rural stations are largely surrounded by natural land cover types (LCZs A–G) and most urban stations are located within built-up areas (LCZs 2–10) with the exception of GEON (LCZ B), UHII is significantly strong in most urban stations, particularly during the day-time. The highest average UHII values were observed in those urban stations located in LCZ 2, LCZ 5, and LCZ 8, where the greatest heat load generally occurs. Our results show that UHII correlates well with sky-view factor, amount of vegetation, and density of built area (although only in the cases of UHII calculated for

daily mean air temperature or daily minima), and, on the contrary, does not show any significant correlation with altitude, similar to the results of previous research in Brno [53].

Interestingly, the UHII and HM values when air temperatures reach their daily minima are relatively low or even negative at some urban sites (e.g., KRAV and ZIDE, both LCZ 9); however, they are located predominantly beyond the city centre and in less dense built-up areas with more vegetation. The night-time UHII has low values during HWs farther from the historical centre, since it is highly dependent on the size of the UHI, which is spatially smaller during the night-time [64]. Furthermore, urban greenery decreases UHII during HWs due to the cooling effect of vegetation cover at night [74]. In addition, according to Richard et al. [46], UHII is generally higher during the first days of HW and then decreases later during HW due to an increase in air temperature in rural areas, while air temperature in the urban environment remains unchanged. This decreasing trend of UHII during HWs may be associated with a reduction in evapotranspiration and soil wetness at the rural site and may be the cause of lower UHII values at night during HWs.

Although the magnitude of UHI is generally higher during the night-time in mid-latitude cities, this is not the case in Brno, given that UHII is significantly higher for daily maxima. Similarly, the contribution of HWs to the UHII is also considerably higher during the day, as our result of the HM analysis shows. However, there is no indication that land use parameters are impacting this increasing day-time UHII and HM, as almost no correlation was found between them. Since the higher HM values during the day in Brno cannot be easily explained by any land use characteristics, other factors are probably more important. This high day-time extremity during HWs can be attributed, for example, to higher surface temperatures [75], enhanced urban–rural differences in surface evapotranspiration [76] or the release of anthropogenic heat from air conditioning [40]. In addition, the results are influenced by a particular small group of urban sites, e.g., GEON (LCZ B) or ZIDE (LCZ 9) which has a great share of vegetation in their close vicinity. However, contrasting forested areas, urban greenery can increase UHII in some parts of the city during the day-time [77], and these stations have warmer air temperature conditions than most other urban stations with a smaller share of vegetation and higher building density.

## 5. Conclusions

In conclusion, our results show that the urban canopy heat island in Brno is considerably strong and effectively amplified by heat waves, particularly during the day-time. Therefore, urban environments are exposed to an even more severe heat load when heat waves influence the local climate of Brno. In addition, heat waves have recently become significantly more frequent, longer, intense, and severe. As a result, the combined effect of increasing heat wave characteristics and more strong intensity of canopy urban heat islands further enhances the heat exposure of the urban population and has a negative impact on the quality of life and even on the health of city residents. Consequently, additional research, which would focus on factors that affect urban climate, can broaden the understanding of the issue, and suggest potential mitigation strategies for city planning management to reduce these negative consequences and improve human thermal comfort in urban areas. In the future, simulations of air temperature conditions can be performed with urban climate models to overcome the lack of spatial information in empirical data from meteorological observations and simulate the prospective benefits of applied mitigation strategies.

**Author Contributions:** Conceptualization, Z.J. and P.D.; methodology, P.D.; software, P.D.; formal analysis, Z.J. and P.D.; investigation, Z.J.; resources, P.D.; data curation, Z.J.; writing—original draft preparation, Z.J.; writing—review, and editing, Z.J. and P.D.; visualization, Z.J.; supervision, P.D. All authors have read and agreed to the published version of the manuscript.

**Funding:** This research was funded by the Ministry of Education, Youth and Sports of the Czech Republic for SustES—Adaptation strategies for sustainable ecosystem services and food security under adverse environmental conditions project ref. CZ.02.1.01/0.0/0.0/16_019/0000797. Z.J. also received funding from Masaryk University within the MUNI/A/1393/2021.

**Data Availability Statement:** The data used in this study can be provided upon request.

**Conflicts of Interest:** The authors declare no conflict of interest.

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
