# Peer review of "Heat Waves Amplify the Urban Canopy Heat Island in Brno, Czechia"

_2674-0494, doi:10.3390/meteorology1040030_

Round 1

Reviewer 1 Report

Authors made interesting research that can be considered for publication in the Meteorology journal, but there are a few recommendations related to minor changes:

1) Introduction, line 71 - I suggest to add this reference, too: https://link.springer.com/article/10.1007/s11069-017-3160-4

2) 2.3 sub-chapter - I do not see any references relate to methodology of land use classification. Did you created the method or you used existing methods from some previous published articles? Please, add reference/s if you used someone methods.

Reviewer 2 Report

This study focused on the heatwave measures and its contribution to heat island effect. The topic is worthy of investigation, and the paper is well written. However, I still have some concerns.

1. Line 39: In a precise expression, canopy UHI is defined as higher air temperature in urban areas than in rural counterparts.

2. Figure 1 lacks the north pole. Did the authors use the same scale bar for three sub-figures? It seems the landscape map of Brno was clipped from Google Earth directly. Please add a reference. It’s better to draw the geographic location map by themselves with land use and land cover types.

3. How did the authors calculate the SVF, BSF and NDVI of each meteorological station? Using buffer analysis? A radius of 300 m for the calculation of all parameters?

4. Did the authors mean there may be missing data within the time-series record of a special-purpose station?

5. How did the authors calculate the average annual number of HW days? It’s better to give exact equations for QHW, DHW, IHW and SHW.

6. Did the authors calculate daytime UHII as difference between urban daily maxima and rural average temperature? Why not using mean temperature during daytime? The same question for nighttime UHII calculation

7.  Please give more explanations about the density curve.

8.  Please elaborate the reason for using Spearman’s rank correlation analysis instead of other correlation analysis methods, e.g., Pearson’s correlation.

9. Figure 2: Please add a legend for (a). What land categories are 18-27? Are class 1-17 LCZ classification scheme?

10. Figure 3: according to (b) and (c), urban-rural differences in DHW and IHW are not evident, particularly for UKZU, BOTA. How can readers get information of the exact years from this figure? Like line 241 “…during the same period…”, line 248 “…hot summers 2015 and 2018 with the longest lasting 15 days from July 27 to August 10, 2018”.

11. Figure 4: For me, the frequency of HW lasting for over ten days is quite similar between two rural stations and several urban stations, e.g., MEND, ZABO, BOTA

12. Line 258, I think DHW didn’t vary widely at urban stations or between urban and rural stations.

12. Please give more details about the calculation procedure of HM. For instance, for station A, is the HM referred to as the difference between average UHII(HW) and average UHII(NHW) during JJA? Have the authors excluded the influence of weather conditions, like wind and precipitation? Since there is a time difference between UHI (HW) and UHI (NHW), they probably have different weather conditions, which may influence the UHI.

13. Would the authors add some discussions about the intra-urban difference in HWs, UHI and HM? I noticed that section 3.1 is entitled “urban-rural and intra-urban differences in HW measures”. However, little analysis of the intra-urban difference was conducted. Notably, the authors adopted the LCZ scheme designed for intra-urban thermal variation analysis.

14. It is not surprising that LCZ 9 shows even negative UHII, which is consistent with many other studies. LCZ 9 (sparsely built) is characterized by large areas of vegetation and several sparsely distributed buildings.

15. Line 4778-483: These statements are quite interesting and inspiring. Could the authors provide data analysis results to support their conclusion that “UHIII is generally higher during the first days of HW and then decreases later during HW”?

16. From my perspective, nighttime UHI is not higher than daytime UHI because the authors used minimum air temperature for calculation. Normally, UHI develops to its maximum magnitude several hours (e.g., two-to-three hours) after sunset during summer. However, the minimum air temperature appears after midnight and before sunrise. The authors didn’t calculate UHII at their potential peak time.
